 | Microbiology **Spectrum**

ⓐ | **Open Peer Review** | Clinical Microbiology | Research Article

# Increase of *Clostridioides difficile* PCR ribotype 002 infection with decreased disease severity in hospitalized patients in the Netherlands

A. Baktash,[1] C. Harmanus,[1,2] J. Goeman,[3] W. Brennan,[4] M. Cormican,[5,6] J. Corver,[1] E. J. Kuijper,[1,2] W. K. Smits[1,2]

**ABSTRACT**  *Clostridioides difficile* PCR ribotype (RT) 002 is the second most prevalent RT in the Netherlands. In 2019, an increase in *Clostridioides difficile* infections (CDIs) due to clonal RT002 strains was reported in Ireland. In retrospect, a significant increase in the proportion of CDI due to RT002 was detected in the Netherlands between May 2017 and May 2018 (11.9% of 683 cases vs 6.5% of 865 in the previous year). Our aim was to analyze and characterize CDI due to RT002 over time and to assess whether RT002 isolates in the Netherlands were related to the Irish RT002 isolates.

During the period from May 2009 to April 2021, Dutch patients with CDI due to RT002 showed less severe CDI (22%) in comparison with hypervirulent RTs (RT027, 29%, and RT078, 25%), similar to grouped data from other non-hypervirulent ribotypes (21%). However, RT002-associated CDI prior to the increased proportion (2009–2017) showed CDI severity similar to CDI associated with hypervirulent types, whereas this was no longer the case after the increase (2017–2021). Using core genome multi-locus sequence typing (cgMLST), Dutch isolates were found not to be related to the Irish outbreak strain. Several Dutch strains clustered together without an evident epidemiological link, suggesting recent emergence. We conclude that the increase of RT002 in Ireland and the Netherlands does not represent clonal spread but is more likely associated with clinical, epidemiological, or treatment changes.

**IMPORTANCE**  *Clostridioides difficile* is a significant cause of hospital-associated infections, making it relevant to understand the emergence and spread of different ribotypes. This study highlights a rise in cases of PCR RT002 in the Netherlands, suggesting changes in its prevalence. Despite the increase in RT002 infections, the severity of disease appears to have decreased and is lower compared to more virulent ribotypes. Further research is needed to understand the factors driving this change. Additionally, this study underscores some limitations of cgMLST. cgMLST is valuable for identifying genetic relationships. However, it cannot always distinguish between widespread, recently emerged strains and local transmission chains, making it challenging to determine the exact sources of infection. Continued surveillance and genetic characterization of *C. difficile* are essential to better understand these trends and improve control measures.

**KEYWORDS**  *Clostridioides difficile* infection (CDI), RT002, core genome MLST, incidence, severe CDI, complicated CDI, molecular epidemiology, sentinel surveillance

*C*lostridioides difficile is an anaerobic, spore-forming, gram-positive bacterium that causes infections varying from antibiotic-associated diarrhea to life-threatening pseudomembranous colitis (1). So-called hypervirulent PCR ribotypes (RTs) such as RT027

**Peer Reviewer** Yoshitomo Morinaga, Toyama Daigaku, Toyama, Japan

Address correspondence to E. J. Kuijper, e.j.kuijper@lumc.nl, or W. K. Smits, w.k.smits@lumc.nl.

E. J. Kuijper and W. K. Smits contributed equally to this article.

The authors declare no conflict of interest.

and RT078 are associated with severe and recurrent *Clostridioides difficile* infections (CDIs), but CDI and outbreaks are not limited to these ribotypes (1).

RT002 is one of the most common ribotypes in the Netherlands and in Europe (2). Besides one case report about *C. difficile* bacteremia due to RT002 in a patient from Belgium (3), little information is available on RT002-associated CDI in Europe, with most publications stemming from East Asia, where RT002 is highly prevalent. *In vitro* assessment of bacteriological characteristics of RT002 strains showed a higher amount of toxin A and toxin B production, sporulation, and germination compared to strains belonging to other ribotypes (e.g., RT012, RT014, and RT046) (4). Comparable results were reported in another study from Hong Kong (5). RT002 also caused more weight loss and histological damage in a murine model of CDI (4). RT002 is highly prevalent among nursing home residents in Hong Kong (5,6), and RT002 infections were associated with high morbidity and mortality in elderly patients (7). Patients with CDI due to RT002 showed more usage of β-lactam antibiotics in the previous 3 months compared with controls (5). Furthermore, RT002 was associated with fluoroquinolone resistance (7).

In 2019, an increase in CDI with a clonal strain of RT002 was reported in Ireland during the third quarter (38% of the ribotyped cases vs 25% in the first quarter) (8). Prompted by this observation and stimulated by the spread of this information by the European Center for Disease Prevention and Control, we assessed the prevalence of CDI due to RT002 in the Netherlands and retrospectively noted an increase in the proportion of CDI due to RT002 between May 2017 and April 2018, compared to previous years (11.9% vs 5.4%–7.2% in the previous years). Furthermore, we assessed whether Dutch RT002 isolates were related to Irish clonal strains. In addition, we compared the clinical characteristics of CDI due to RT002 with those of CDI caused by both hypervirulent and non-hypervirulent ribotypes.

Considering an increase in the proportion of CDI due to RT002 in the Netherlands in 2017, we also performed a sub-group analysis in which the strains were grouped based on isolation date.

## MATERIALS AND METHODS

### Clinical data collection

Clinical data collected from May 2009 until April 2021 ($n$ = 8,080 strains PCR-ribotyped) from the Dutch National CDI Sentinel Surveillance were used to analyze the clinical characteristics of CDI episodes due to RT002. All hospitalized patients (>2 years old) in participating hospitals (N = 24 acute care hospitals) with clinical signs of CDI, in combination with a positive test for *C. difficile* toxins or toxigenic *C. difficile*, were registered. The local laboratory chose the indication for testing for CDI, the assay, and the algorithm that were used for diagnosis of CDI. CDI was described as severe if at least one of the following conditions was present: fever (≥38°C) and leukocytosis (>15 × 10$^9$ cells/L), dehydration and/or hypoalbuminemia (<20 g/L), bloody diarrhea, and pseudomembranous colitis (9). A complicated CDI course was defined as admission to the intensive care unit and the need for a surgical procedure and/or mortality within 30 days after diagnosis of CDI (either due to CDI or non-CDI related) (9).

Clinical characteristics of CDI episodes due to RT002 were compared with the results of other RT groups. These groups were RT027 and RT078/RT126 (which are considered hypervirulent strains), RT001 and RT014/RT020/RT295 (which are common non-hypervirulent strains in the Netherlands), and the other group (excluding the four previous strains). Ribotypes RT014/RT020/RT295 and RT078/RT126, which were difficult to distinguish from each other with PCR ribotyping, but not with capillary electrophoresis PCR ribotyping, were combined in one group, since the data collecting period encompassed both methods. The data set was also split in two based on isolation date.

## Sequence data

We sequenced 39 randomly selected strains of RT002 collected between 2013 and 2021. The metadata can be found in Table S1 in the supplementary files. Total DNA was isolated from cultured bacteria for sequencing these strains. In short, RT002 colonies were resuspended in Tris/EDTA buffer and heated at 100°C for 10 min according to the protocol (10). Chromosomal DNA was isolated using the QiaAmp blood and tissue kit (Qiagen) according to the instructions of the manufacturer. After preparation with the NebNext Ultra II DNA library prep kit for Illumina, DNA was sequenced at GenomeScan B.V. (Leiden, The Netherlands) on an Illumina NovaSeq 6000. On average, 3 million paired-end reads (read length 150 bp) were produced per sample, containing a minimum of 90% reads with a quality of 30 or more.

## Ridom SeqSphere⁺ genome multi-locus sequence typing

Ridom SeqSphere⁺ (version 6.0.2; Ridom GmbH, Münster, Germany) was run with default settings for quality trimming, *de novo* assembly, and allele calling. Quality trimming occurred at both 5′ and 3′ ends until reaching an average base quality of 30 with a length of 20 bases and a 120-fold coverage (11). *De novo* assembly was performed using the SKESA assembler (version 2.3.0) (12) integrated in SeqSphere⁺, applying default settings. SeqSphere⁺ scanned for the defined genes using BLAST (13) with the criteria described previously (11).

Quality control of these assemblies was performed. The average assembly size was 4.2 Mb. The average N50 across the samples was 213,448 bp (range: 18,044–501,411 bp). The guanine-cytosine (GC) content was consistent at 28.21% across samples. The average contig size was 106, with a range of 57–502 contigs. Core genome multi-locus sequence typing (cgMLST) analysis showed an average of 99.75% correctly assembled core loci. One sample formed an outlier with an average assembly size of 7.3 Mb and a GC content of 30.6%, indicating possible contamination. However, the cgMLST analysis showed 99.7% correctly assembled core genome loci. Despite minor variations, the overall assembly quality was high and suitable for downstream analyses.

For further analysis, distance matrices, minimum spanning trees (MSTs), and neighbor-joining trees (NJTs) were produced using the integrated features within SeqSphere⁺ using the "pairwise ignoring missing values" option. To verify the result of the NJT, whole genomes were compared to one another using average nucleotide identity with fastANI (version 1.33) (14).

## Statistical analysis

Age was compared by a Wilcoxon rank-sum test, and all other characteristics were compared by Pearson's chi-square test, and in case of expected frequencies of <5, Fisher's exact test was used. To compare the effect of RT002 and other ribotypes on clinical characteristics, separate multiple logistic regressions were performed on data restricted to only RT002 and the ribotype of interest, with age, antibiotic usage, and RT as independent variables. To account for multiple hypothesis testing, the Bonferroni correction was applied specifically to the analyses involving the variables severe CDI and CDI-related mortality. This correction adjusts the significance threshold ($a$) by dividing the original significance level (set at 0.05) by the total number of independent tests conducted for these variables. A result was considered statistically significant only if the *P* value was less than the adjusted $a$; for Table 1, α = 0.005 (10 tests were included), and for Table 2, α = 0.0125 (4 tests were included). The statistical analysis was performed on IBM SPSS Statistics for Windows (version: 28.0.1.0) (IBM Corp., Armonk, NY, USA).

**TABLE 1** Comparison of clinical characteristics of patients with RT002 with other RTs (2009–2021)[a]

| | RT002 (n = 611) | Other RTs (n = 4,145) | RT027 (n = 125) | RT078/RT126 (n = 960) | RT014/RT020/RT295 (n = 1,396) | RT001 (n = 843) |
|---|---|---|---|---|---|---|
| Age | 72.62 (58.19–82.04) | 70.40 (58.07–79.83) | 73.77 (61.48–81.27) | 71.15 (61.22–80.44) | 70.42 (58.59–79.74) | 75.06 (63.74–82.66) [b]P < 0.001 |
| Men | 298/611 (48.77) | 2,039/4,145 (49.19) | 67/125 (53.60) | 482/960 (50.21) | 670/1,396 (47.99) | 408/843 (48.40) |
| Severe CDI | 128/584 (21.92) | 850/3,908 (21.75) | 32/111 (28.83) | 242/910 (26.59) [c]P = 0.011 | 256/1,322 (19.36) | 134/789 (16.98) |
| Complicated course | 76/540 (14.07) | 442/3,544 (12.47) | 22/100 (22.00) [c]P = 0.006 | 123/811 (15.17) | 115/1,201 (9.58) [c]P = 0.033 | 110/697 (15.78) |
| CDI mortality | 18/540 (3.33) | 88/3,544 (2.48) | 5/100 (5.00) | 30/811 (3.70) | 22/1,201 (1.83) | 30/697 (4.30) |

[a]Data are shown as no. of cases (%) or, for age, as median (Q1–Q3) due to skewed distribution. The "other RTs" group includes all ribotypes except RT001, RT014/RT020/RT295, RT027, and RT078/RT126.
[b]P < 0.05 after correction for antibiotic use.
[c]P < 0.05 after correction for age and antibiotic use. An adjusted alpha level (α = 0.005) was used for severe CDI and CDI-related mortality. The supplementary material includes a detailed table of clinical characteristics and a combined heatmap visualizing the pattern of these characteristics across the RT groups.

## RESULTS

### CDI in hospitalized patients due to RT002 is comparable with other non-hypervirulent ribotypes

Following the reported RT002 rise in Ireland (8), we analyzed data from the Dutch National Expertise Center for *C. difficile* infections and observed a significant increase in RT002-related CDI from 5.4% to 7.2% (May 2009–April 2017) to 11.9% (between May 2017 and April 2018) (Fig. 1). The proportion of CDI due to RT002 remained between 9.6% and 9.9% in the years following this period (May 2019–April 2021). In order to ascertain if this increase in proportion was accompanied by changes in clinical characteristics of patients, we used the data provided by the Dutch national *C. difficile* surveillance program between May 2009 and April 2021 using 8,080 PCR-ribotyped isolates from hospitalized patients in 24 hospitals in our analysis. Within this set, the RT002 group accounted for 611 isolates. The average proportion of RT002 was 7.5%, making it the third most common RT behind RT014/RT020/RT295 (17.3%) and RT001 (10.3%) in the Netherlands during that period.

When demographic data, clinical characteristics, and 30-day outcome of patients with CDI due to RT002 (2009–2021) was compared with CDI due to other ribotypes (Table 1 and Table S2 in the supplementary file), there was no significant difference in age or gender, except for the RT001 group, which had a higher median age, even when corrected for antibiotic usage (P < 0.001). The RT002 group showed a higher severe CDI average (21.92%) compared to the RT001 group (16.98%, P = 0.021) but not significant when corrected for age and antibiotic usage (P = 0.158). However, the RT002 group showed a significantly lower severe CDI average when compared to the RT078/RT126 group (26.59%, P = 0.041), even when correcting for age and antibiotic usage (P = 0.011), but not after correcting for multiple testing. The RT002 group showed a lower complicated course average (14.07%) than the RT027 group (22.00%, P = 0.043); this

**TABLE 2** Comparison of clinical characteristics of patients between two groups of RT002[a,c]

| | RT002: 2009–2017 (n = 348) | RT002: 2017–2021 (n = 263) |
|---|---|---|
| Age | 72.14 (57.93–81.56) | 72.77 (59.11–82.15) |
| Men | 155/348 (44.54) | 143/263 (54.37) P = 0.015[b] |
| Severe CDI | 88/325 (27.08) | 40/259 (15.44) P < 0.001[b] |
| Complicated course | 48/290 (16.55) | 28/250 (11.20) |
| CDI mortality | 13/290 (4.48) | 5/250 (2.00) |

[a]Data are shown as no. of cases (%) or, for age, as median (Q1–Q3) due to skewed distribution.
[b]P < 0.05 after correction for antibiotic use.
[c]P < 0.05 after correction for age and antibiotic use. An adjusted alpha level (α = 0.005) was used for severe CDI and CDI-related mortality. The supplementary material includes a detailed table of clinical characteristics and a combined heatmap visualizing the pattern of these characteristics across the RT groups.

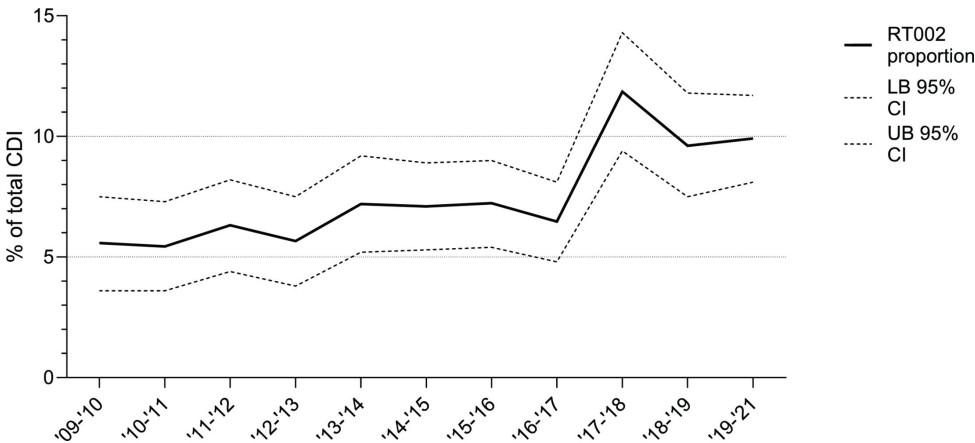

**FIG 1** Proportion of RT002 of the total CDI in time in the National Sentinel Surveillance samples. The lower boundary (LB) and upper boundary (UB) of the 95% CI (CI) are shown with a dotted line.

remained significant after correcting for age and antibiotic usage but disappeared after correcting for multiple testing. The RT002 group showed a higher complicated course average (14.07%, $P = 0.005$; after correcting for age and antibiotic usage, $P = 0.033$) than the RT014/RT020/RT295 group (9.58%).

In Table S2 (the detailed version of Table 1), comparisons of subsets of severe CDI (dehydration and/or hypoalbuminemia, bloody diarrhea, pseudomembranous colitis, and fever and leukocytosis) are made between RT002 and other RTs. Furthermore, overall mortality, onset of CDI, recurrent CDI, and antibiotic usage are compared between these groups. RT002 had a higher overall mortality average (12.96%, $P = 0.004$; after correction for age and antibiotic usage, $P = 0.034$) than the RT014/RT020/RT295 group (8.49%). However, the CDI-related mortality did not differ. The onset of symptoms of CDI in the RT002 group (42.39%) was more frequent at home compared with the RT027 group (31.82%, $P = 0.003$; after correction for age and antibiotic usage, $P = 0.108$) and the RT001 group (26.21%, $P < 0.001$, unchanged after correction). Antibiotic usage prior to CDI was significantly lower in the RT002 group (63.64%), compared with the RT001 group (76.81%, $P < 0.001$, also after correction for age), the RT027 group (78.10%, $P = 0.004$; after correction for age, $P = 0.006$) and the RT078/RT126 group (71.50%, $P = 0.002$, also after correction for age). Overall, CDI due to RT002 is comparable to other non-hypervirulent ribotypes and occurs more frequently at home and is associated with less antibiotic usage.

## CDI in hospitalized patients due to RT002 in May 2009–April 2017 was severe and comparable with CDI due to RT027 and RT078/126 strains

Considering the proportion increase of RT002 in 2017, we next split the data set in RT groups prior to the proportion increase (hereafter RT$_{09-17}$, May 2009–April 2017) and in RT groups after this increase in proportion (hereafter RT$_{17-21}$, May 2017–April 2021) to assess whether there were changes in demographics, clinical characteristics, and 30-day outcome between the two temporal groups.

The RT002$_{09-17}$ group and the RT002$_{17-21}$ group accounted for 348 and 263 isolates, respectively. We observed a significant difference between the RT002$_{09-17}$ group (Table 2 and Table S3 in the supplementary file) in severe CDI (27.08% vs 15.44%, $P < 0.001$, also after correction for age and antibiotic usage) and dehydration and/or hypoalbuminemia average (16.62% vs 4.59%, $P < 0.001$, also after correction for age and antibiotic usage) and lower incidence in males (44.54% vs 54.37%, $P = 0.018$, after correction for age and antibiotic usage $P = 0.015$) compared to the RT002$_{17-21}$ group.

We noted that the characteristics of the RT002 isolates relative to the other RT groups appeared to differ between May 2009 and April 2017 and between May 2017 and April

2021 (Table S4A and B). In particular, RT002 was more similar to hypervirulent RTs (RT027 and RT078/RT126) prior to May 2017, whereas it was more similar to non-hypervirulent RTs during the period from 2017 to 2021. The $RT002_{09-17}$ group (Table S4A) showed a significantly higher rate of severe CDI, fever, and leukocytosis and complicated CDI course than non-hypervirulent RTs and a higher rate of dehydration and/or hypoalbuminemia than non-hypervirulent RTs and the other RTs group. In contrast, the $RT002_{17-21}$ group (Table S4B) had a comparable severe CDI average as the non-hypervirulent RTs and the other RT group, but lower compared to hypervirulent RTs (e.g., $RT078/RT126_{17-21}$ group; 15.44% vs 25.00%, $P = 0.006$; also after correction for age and antibiotic use, $P = 0.003$). The average complicated CDI course of the $RT002_{17-21}$ group was comparable with all ribotypes.

When other ribotypes were analyzed in the same temporal subgroups, we did not observe such a trend (Table S5 through S7). For instance, both $RT001_{09-17}$ and $RT014/RT020/RT295_{09-17}$ appeared to cause less severe disease than other ribotypes in the same period, whereas this was increased for $RT002_{09-17}$.

## Dutch RT002 strains are not related to the Irish outbreak strains, but certain Dutch strains are genetically related to each other

As the increase in proportion of CDI due to RT002 in the Netherlands appeared to precede the Irish reports, we investigated whether the Irish clonal strains were related to the Dutch RT002 strains. Thirty-nine randomly selected RT002 strains from the Netherlands between 2013 and 2021 were sequenced and compared to sequence data from 11 Irish strains (collected in 2019 and 2020), consisting of 6 Irish RT002 outbreak strains and 5 randomly selected RT002 strains (see Table S1 for the metadata).

A minimum spanning tree was made based on cgMLST using SeqSphere$^+$ to assess clustering of strains (Fig. 2A), and a neighbor-joining tree is shown to assess the phylogenetic relationship of the Dutch strains with the Irish outbreak strains (Fig. 2B). The Dutch RT002 strains showed no relatedness with the Irish outbreak strains in the cgMLST analysis. Five clusters were detected with cgMLST (Fig. 2A), one of which corresponded to the Irish clonal outbreak isolates that were included as controls in our analysis (MST Cluster 1). These outbreak strains (Irish 6–11) formed a tight cluster with 0 allele difference in a cgMLST analysis. All Dutch RT002 isolates had >6 allele differences to these isolates. Two Dutch strains showed 9 allele differences with cgMLST analysis. One Irish non-outbreak (Irish 5) strain was related to MST Cluster 1 and differed at three alleles. The other non-outbreak Irish isolates (Irish 1–4) were interspersed in the minimal spanning tree.

The strains from MST Cluster 2 differed by three to six alleles from each other in a distance matrix (data not shown). MST Cluster 2 consisted of five strains, of which three were obtained from hospitalized patients from one hospital across different years (2013, 2017, and 2018). All three patients had the onset of their CDI outside the hospital: one patient in a long-term residential care facility and the other two at home. The other two strains were from two different hospitals at a distance of 50 and 110 km from the first three. One patient developed CDI at home and the other in the hospital. All five patients lived in a different city within a radius of 55 km from each other. All isolates were from patients with mild CDI with an uncomplicated CDI course. Due to the close relatedness of MST Cluster 2 and MST Cluster 3, strain 2018-5 from MST Cluster 2 is shown separately from the other strains of MST Cluster 2 in the NJT (Fig. 2B). This has also been verified using FastANI, showing high pairwise average nucleotide identity between strain 2018-5 and strains from Clusters 2 and 3 (ranging from 99.845% to 99.992%, heatmap not shown).

Cluster 3 was genetically related to MST Cluster 2, at a minimum distance of 8 allele differences. The strains from MST Cluster 3 were from three different hospitals in different cities. The strains from MST Cluster 3 differed by one to five alleles from each other. This group consisted of two mild CDI cases and one severe CDI case; all had an uncomplicated CDI course and developed CDI at home. All these patients lived in different cities within a radius of 60 km.

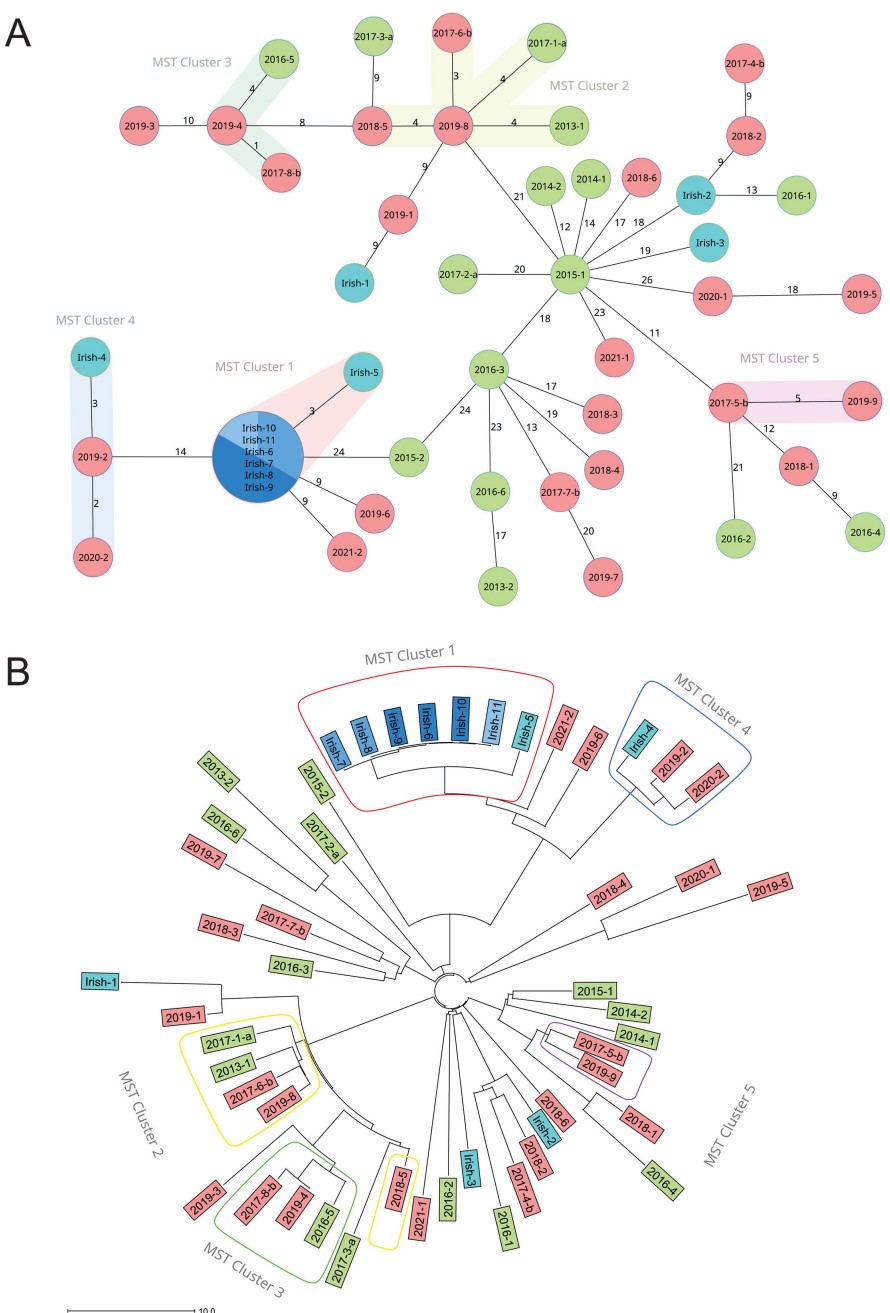

**FIG 2** Analysis of relatedness of RT002 strains in this study. (**A**) SeqSphere[+] cgMLST analysis with minimum-spanning trees of Irish RT002 random strains ($n = 5$, Irish 1–5, depicted as light blue) and clonal strains ($n = 6$, Irish 6–11, depicted as several shades of blue inside of the biggest node) and random Dutch RT002 strains ($n = 39$, RT002$_{09–17}$ strains depicted as green and RT002 $_{17–21}$ strains depicted as red). Dutch strains are stated as "year of isolation-sample." Strains from 2017 end with "a" or "b," indicating strains prior or after the increase in proportion, respectively. One or more strains inside a node means that these strains have 0 allele difference. The colored septations represent the laboratory from where the strains have been obtained. The numbers between each node correspond to the number of different alleles between the strains. The colored shadowing of circles represents a cluster with at most 6 allele differences that are genetically related. (**B**) Neighbor-joining tree based on SeqSphere[+] cgMLST allele difference from 50 strains of RT002 also shown in Fig. 2. The distance is given as the absolute allelic difference. The colored encircling of strains represents a cluster with at most 6 allele differences that are genetically related: Cluster 1, red shade; Cluster 2, yellow shade; Cluster 3, green shade; Cluster 4, blue shade; Cluster 5, purple shade.

MST Cluster 4 consisted of two Dutch strains from different cities and one Irish non-outbreak strain (Irish 4). The allele differences in these strains were two to four alleles. One patient had a mild CDI course, and the CDI course of the others was not known.

Lastly, MST Cluster 5 consisted of two strains from different cities at a distance of 270 km, with an allele difference of 5. These isolates were also from patients with a mild CDI course and developed CDI at home. In Fig. 2B, a neighbor-joining tree is shown with the phylogenetic relationship of the Dutch strains with the Irish outbreak strains and where all clusters can be seen.

## DISCUSSION

Our main goal was to study the increase in proportion of CDI due to RT002 in hospitalized patients to assess whether this was related to the Irish RT002 outbreak and whether this was accompanied by a change in clinical characteristics. We showed using cgMLST that the sequenced Dutch RT002 strains are not genetically related to the Irish outbreak strains. However, we found several clusters of RT002 strains in the Netherlands, suggesting clonal expansion. Though some of these clusters were recognized in the same hospital, most of these clusters comprised strains from different geographic locations at different time points, arguing against an outbreak or local transmission. This was also seen with the Irish outbreak strains, which showed a cluster with cgMLST (0 allele difference) but derived from different laboratories in different counties (two from known counties and three known laboratories and one unknown county and laboratory).

The overall CDI incidence was on average 3.06 cases per 10,000 patient-days between 2010 and 2021, and CDI incidence due to RT002 was on average 0.22 cases per 10,000 patient-days. The increase in proportion of RT002 that was noticed between May 2016 and April 2017 and between May 2017 and April 2021 (Fig. 1) preceded a general increased CDI incidence 1 year later (Fig. S1).

The increase of CDI due to RT002 was not accompanied by an increase in severe disease. The clinical characteristics of hospitalized patients with CDI due to RT002 were similar to other non-hypervirulent ribotypes when assessed for the full 2009–2021 period. However, we observed that RT002 strains showed a higher CDI severity before the increased proportion, together with a higher complicated course and higher mortality. This was not observed in other ribotypes that were analyzed in the same temporal subgroups.

The reason for the increased proportion in combination with a decrease in disease severity in RT002 requires further research. Several microbiological causes for an increased proportion can be envisaged, such as mutations or acquisition of traits that confer advantages (such as differences in metabolism, altered toxin expression, antibiotic resistance, or enhanced survival in specific environments). A preliminary *in silico* analysis on the sequenced RT002 strains ($n$ = 15 from May 2009 to April 2017 and $n$ = 25 from May 2017 to April 2021, Table S1) did not reveal alterations in or acquisition of genes associated with trehalose metabolism (15) or changes in the pathogenicity locus associated with increased virulence that were associated exclusively with one group or the other (16). Though a VanS T349I mutation in RT027 has been associated with an increase in vancomycin MIC (17), antimicrobial susceptibility testing of RT002 strains carrying this mutation did not show elevated vancomycin MICs, and the MIC did not differ between strains in 2016–2017 and 2017–2019 (MIC <0.06 mg/L). In addition, no resistance determinants or phenotypic reduced susceptibility was found for metronidazole (MIC in both aforementioned time periods 0.125 mg/L). The higher rates of sporulation and germination of RT002 reported previously (5,6) may possibly account for persistence of RT002 strains in the environment and could imply that its spores could germinate even in suboptimal environments. This could contribute to an increased proportion of RT002 CDI cases. However, these differences in proportion of CDI with changing severity might also result from non-microbiological factors, including dietary habits, change in infection prevention measurements, and host factors. It is possible

that affected patients differ in physical condition and comorbidities before and after 2017. We were unable to collect data to analyze the McCabe score, though this score is present in our current national surveillance program since 2019. The first-line antibiotic treatment for patients with CDI was metronidazole and changed in 2023 when national guidelines preferred vancomycin and fidaxomicin; thus, it is unlikely that a difference in CDI treatment contributes to our observations.

We found clustering of strains belonging to *C. difficile* RT002 based on a cgMLST analysis using SeqSphere[+], which uses ≤6 allele differences to define an MST cluster. Because the strains were randomly selected, we did not apply the ≤3 allele difference threshold proposed for outbreak settings in our previous study (18); moreover, even with this stricter threshold, distinguishing outbreak from non-outbreak strains remains challenging within certain MLST clades (18). The average allele difference between RT002 strains from within and outside the Netherlands was 28.5 and 32.5 allele differences, respectively, in our previous study. Furthermore, strains from MLST Clade 2 had a lower intra-ribotype allele average (9 alleles) compared to MLST Clade 1 (114 alleles), to which RT002 belongs. Therefore, clonality should be easier to detect in comparison with a ribotype belonging to MLST Clade 2. For instance, we found a cluster (see Fig. 2A, Cluster 2) comprising three patients with community-onset CDI hospitalized with CDI due to RT002 (3–6 allele differences) in different years in the same hospital. It is possible that RT002 spores persist for several years in the hospital environment and occasionally infect patients who subsequently develop CDI after discharge. Another better explanation could be a yet unknown common source outside the hospital. For instance, a study in 12 European countries showed that various PCR ribotypes, including RT002, were present on retail potatoes (19). The most likely explanation is that the data from cgMLST for RT002 are limited for patient transmission studies.

During the preparation of this paper, a study on the genetic characteristics of RT002/sequence type (ST) 8 was published (20). The authors observed close genetic relatedness among the studied ST8 genomes from different European countries, acquisition of a varied array of antimicrobial resistance, and presence of numerous mobile elements. Furthermore, they found clonal isolates across distinct One Health sectors, geographic location, and time periods. These results suggest that isolates of RT002 are genetically more closely related than other RT lineages and that related isolates can be detected independently from the geographic origin. Clonality (e.g. in terms of cgSNP differences) therefore does not necessarily indicate a direct transmission.

The strengths of this study are the 12 years of available data from hospitals in different geographic regions that participated in the sentinel CDI surveillance and the high sample size.

There are also a few limitations. Our study relies on retrospective data, which can be affected by incomplete records and reporting bias. The relatively small number of sequenced strains and uncommon ribotype groups with a low incidence after splitting the data set in two groups (e.g., PCR ribotype 027) limits statistical power. The fact that only strains from the Netherlands and Ireland were included here may limit the generalizability of our results. The small sample size of certain groups limits the number of confounders (e.g., age, antibiotic usage, comorbidities, and treatment) which could be corrected. For other confounders such as comorbidities, the national CDI surveillance only started collecting data since 2019, so we could not correct for comorbidities. Therefore, we only corrected for age and antibiotic usage as a confounder. Only the location of CDI symptom onset and not the location of *C. difficile* acquisition was recorded. However, CDI-attributable mortality is included in the surveillance and data analysis. Due to the absence of patient-level epidemiological data, retrospective interpretation of possible epidemiologically linked clusters is difficult, and transmission routes cannot reliably be inferred (e.g., three strains from MST Cluster 2 from the same hospital).

## Conclusion

This paper highlights that the results of a cgMLST should be interpreted with care with respect to understanding pathways of patient transmission. Without the context of wider sequencing data, epidemiological data, and metadata, cgMLST cannot differentiate between links attributed to widely disseminated clonal strains (e.g., in food or water) and those related to site-specific chains of transmission of infection (local outbreaks) (18). This differentiation is essential for appropriately targeted interventions. The usage of cgMLST, in combination with a national sentinel surveillance system, is important to detect the emergence of certain strains over the years. It is, therefore, of great importance to continue epidemiological surveillance and genetic characterization of *C. difficile.*

## AUTHOR AFFILIATIONS

[1]Leiden University Center of Infectious Diseases (LUCID), Leiden University Medical Center, Leiden, the Netherlands

[2]Dutch National Expertise Centre for *Clostridioides difficile* Infections, National Institute of Public Health and the Environment, Leiden University Medical Center, Leiden, the Netherlands

[3]Department of Biomedical Data Sciences, Leiden University Medical Centre, Leiden, the Netherlands

[4]National Carbapenemase-Producing Enterobacterales Reference Laboratory and National Salmonella, Shigella and Listeria Reference Laboratory, University Hospital Galway, Galway, Ireland

[5]Division of Medical Microbiology, Galway University Hospital, Galway, Ireland

[6]Discipline of Bacteriology, College of Medicine, Nursing and Health Sciences, University of Galway, Galway, Ireland

## AUTHOR ORCIDs

M. Cormican http://orcid.org/0000-0002-8023-1882
J. Corver http://orcid.org/0000-0002-9262-5475
E. J. Kuijper http://orcid.org/0000-0001-5726-2405
W. K. Smits http://orcid.org/0000-0002-7409-2847

## AUTHOR CONTRIBUTIONS

A. Baktash, Formal analysis, Investigation, Methodology, Visualization, Writing – original draft, Writing – review and editing | C. Harmanus, Investigation | J. Goeman, Formal analysis, Methodology | W. Brennan, Investigation, Resources, Writing – review and editing | M. Cormican, Investigation, Resources, Writing – review and editing | J. Corver, Formal analysis, Investigation, Supervision, Writing – original draft, Writing – review and editing | E. J. Kuijper, Conceptualization, Supervision, Writing – original draft, Writing – review and editing | W. K. Smits, Resources, Supervision, Visualization, Writing – original draft, Writing – review and editing

## DATA AVAILABILITY

All sequence reads were uploaded and shared on NCBI's Sequence Read Archive (SRA) under BioProject accession no. PRJNA1238352.

## ETHICS APPROVAL

This observational study used pseudonymized data from the Dutch National CDI Sentinel Surveillance program, which has been in operation since 2009. The program collects microbiological and clinical data from all hospitalized patients diagnosed with CDI at participating hospitals across the Netherlands and was developed by the National Institute for Public Health. No additional data, isolates, or materials were specifically collected for this study nor were any actions required from patients.

## ADDITIONAL FILES

The following material is available online.

### Supplemental Material

**Fig. S1 (Spectrum01446-25 s0001.pdf).** Annual incidence of RT002 CDI.
**Supplemental tables (Spectrum01446-25 s0002.xlsx).** Tables S1 to S7.

### Open Peer Review

**PEER REVIEW HISTORY (review-history.pdf).** An accounting of the reviewer comments and feedback.

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
