## [Reviewer comments · Microbiology Spectrum]

Microbiology Spectrum

Increase of *Clostridioides difficile* PCR ribotype 002 infection with decreased disease severity in hospitalized patients in the Netherlands

Amoe Baktash, Celine Harmanus, Jelle Goeman, Wendy Brennan, Martin Cormican, Jeroen Corver, Ed Kuijper, and Wiep Klaas Smits

Corresponding Author(s): Wiep Klaas Smits, Leids Universitair Medisch Centrum

Review Timeline:

Submission Date:	May 9, 2025
Editorial Decision:	June 6, 2025
Revision Received:	June 24, 2025
Editorial Decision:	July 11, 2025
Revision Received:	July 15, 2025
Accepted:	July 22, 2025

Editor: Yuan Pin Hung

Reviewer(s): Disclosure of reviewer identity is with reference to reviewer comments included in decision letter(s). The following individuals involved in review of your submission have agreed to reveal their identity: Yoshitomo Morinaga (Reviewer #2)

Transaction Report:

DOI: <https://doi.org/10.1128/spectrum.01446-25>

Re: Spectrum01446-25 (Increase of *Clostridioides difficile* PCR ribotype 002 infection coincides with decreased disease severity in hospitalized patients in the Netherlands)

Dear Dr. Wiep Klaas Smits:

I have received the reviews of your manuscript and regret to inform you that we will not be able to publish it in *Microbiology Spectrum*. Your submission was read by reviewers with expertise in the area addressed in your study and it was the consensus view of these reviewers that your paper did not meet the standards necessary for publication.

I am sorry to convey a negative decision on this occasion, but I hope that the enclosed reviews are useful. Please note, rejections from *Microbiology Spectrum* are final and your manuscript will not be considered by other ASM journals. We wish you well in publishing this report in another journal and hope that you will consider *Spectrum* in the future.

Sincerely,
Yuan Pin Hung
Editor, *Microbiology Spectrum*

Reviewer #2 (Comments for the Author):

This manuscript provides a valuable genetic analysis of the epidemiology of *Clostridioides difficile*. The study is strengthened by both an adequate survey period and a sufficient number of isolates analyzed, which together enhance the scientific significance of the presented data. The detection of an increase in ribotype 002 is a particularly important finding and stands out as a key observation in this work. However, the academic value of the manuscript could be further elevated if the authors were able to present data suggesting potential factors underlying this increase. Elucidating such factors would provide deeper insight into the observed epidemiological trends.

The identification of an association between RT002 and disease severity is noteworthy, as it highlights an important issue. Nevertheless, the genetic analyses do not yet provide in-depth insights into the pathogenic characteristics of the strains. While the volume of data suggests that the study has potential as a surveillance report, the current manuscript lacks sufficient figures and tables to fully support this approach. As such, readers interested in utilizing the epidemiological data may find the presentation insufficient.

Fundamentally, this is an epidemiological study enhanced by genetic analysis. In terms of both the title and the overall narrative, the relationship between RT002 and disease severity does not appear to be the main focus, but rather a secondary finding that emerged from the broader analysis. This positioning seems appropriate given the current structure and content of the manuscript.

Reviewer #3 (Comments for the Author):

Phenotypic antimicrobial resistance results should be included. These results may also be able to characterize RT002 from 2009 to 2017 and RT002 from 2017 to 2021.

Reviewer #4 (Comments for the Author):

This is an interesting retrospective study relating the epidemiology of CDI caused by RT 002 strains in the Netherlands over a very long observational period. The authors analysed a very large dataset comprising patient data from 2009 to 2021. Out of 8080 strains confirmed to be RT 002 by PCR ribotyping, 39 strains were selected at random. These strains were fully sequenced and compared by MLST methods from the obtained genome sequences. These genome sequences have been uploaded to NCBI and will be an asset to the *C. difficile* research community.

The authors identified two distinct time periods in which the earlier period (2009-2017) corresponded with increased CDI severity that was more comparable to hypervirulent strains than strains of normal virulence. In the later period (2017-2021) although the incidence of CDI ascribed to RT 002 in NL was increased, the severity of CDI was more comparable to strains of standard virulence and less comparable to infection caused by hypervirulent ribotypes. This is an interesting observation that could not be

explained by analysis of the WGS from strains isolated in the earlier and later periods.

Finally, the RT 002 strains isolated in NL during the surveillance period were shown to be genetically distinct from those isolated during a previous outbreak in Ireland. Clonal relatedness was however observed across the various testing sites in NL.

This study provides insight into infection caused by RT 002 *C. difficile* in NL and the changing epidemiology observed across a long period of observation.

Major comments

I have reviewed this paper after the initial round of peer review. From my analysis of the review comments and author responses from the first round, the comments to all 3 reviewers are addressed satisfactorily, in my opinion.

Difference between RT002 in May 2009-April 2017 and May 2017- April 2021 strains. This is very interesting. When discussing that no obvious mutations relating to virulence or sporulation were observed in the conclusions and discussion, please state the number of sequenced strains that correspond to the earlier time period (more severe CDI) and later time period (less severe CDI).

Line 376-377 - It is not clear from this statement whether any genetic differences were observed between these two groups within the conserved PaLoc region at all. The current statement claims that no mutations associated with virulence were detected. But any mutation in this region could have an impact. Please clarify this point.

minor comments

Line 66-67 Abbreviations introduced in the abstract (CDI, RT) should be reintroduced here.

Line 122- define EC-PCR Ribotyping

Line 201 - "We were wondering" is quite informal. Please rephrase more formally e.g. In order to ascertain whether

Line 247 - This title is a little confusing at this step of the manuscript. Consider replacing "proportion increase" with the actual dates: CDI in hospitalized patients due to RT002 in May 2009-April 2017 was severe, and comparable with RT027 and RT078/126 strain

Line 280 - Should this title read "Dutch RT002 strains ARE not related to the Irish outbreak strains, but certain Dutch strains..."

Line 376-377 - Please reference source papers relating to mutations in the PaLoc that affect virulence, rather than the selected review.

Increase of *Clostridioides difficile* PCR ribotype 002 infection with decreased disease severity in hospitalized patients in the Netherlands

Comments:

Phenotypic antimicrobial resistance results should be included. These results may also be able to characterize RT002 from 2009 to 2017 and RT002 from 2017 to 2021.

Reviewer #1 (Major Comments for the Author):

Baktash et al. investigate the epidemiological trends and clinical characteristics of *Clostridioides difficile* ribotype RT002 infections among hospitalized patients in the Netherlands over a 12-year period. The authors report a significant increase in RT002 infections, which coincides with a decrease in disease severity, compared to hypervirulent ribotypes RT027 and RT078.

1. The analysis primarily adjusts for age as a confounder. However, other potential confounders, such as comorbidities and antibiotic usage patterns throughout the study period, are not accounted for. These factors could significantly influence the severity of *C. difficile* infections.

We agree with the reviewer that comorbidities and antibiotic usage throughout the study period could confound the clinical results. We have corrected for antibiotic usage as a confounder. However, it was not possible to correct for comorbidities, since the McCabe score was only used since 2019. To acknowledge the possible confounders, we have described this more clearly in the discussion (L391-396).

2. No correction for multiple testing was performed. While the authors justified this decision by focusing on general patterns, the extensive number of comparisons increases the risk of Type I errors (false positives).

We agree with the reviewer that there is a risk of Type 1 errors due to the extensive numbers of comparisons. We have corrected for multiple testing for the covariates severe CDI and CDI mortality and determined that the conclusions were still valid.

3. Tables 1 and 2 use heatmaps for data visualization. This approach makes it difficult for readers to interpret the precise data. Heatmaps would be better suited as supplementary material, in my opinion.

We thank the reviewer for the feedback. The heatmaps were generated in order to show that trends in categories that are not statistically different support the overall image of a change in disease severity. We accept the reviewer's opinion that this non-standard visualization may not be intuitive for readers used to look at numerical value. We have followed the recommendation, and placed the heatmaps in Supplemental Material, and the numerical tables in the main body of the manuscript. We adjusted the text to reflect this change.

Reviewer #1 (Minor Comments for Author (Required)):

1. The manuscript mentions an increase in RT002 cases in Ireland within the abstract and results sections. However, no references are provided, and no results are shown.

We have now included references to support the statement. Enhanced Surveillance of Clostridioides (Clostridium) difficile Infection. National Report Q3 2019. Ireland: HSE-HPSC; 2019).

2. Why was such an extended timeframe (12 years) chosen when the focus appears to be on a relatively recent increase in RT002 infections?

The increase in the proportion of RT002 strains was noted in 2017, and though RT002 is a common ribotype, it constitutes between 5.4-11.9% of all strains in the Netherlands (as shown in Figure 1). In order to have sufficient numbers of strains to allow for a useful comparison between both periods, we extended the analysis across multiple years.

3. In the heatmaps presented, certain cells are outlined with black borders, while others are not.

We have moved the heatmap tables to the supplementary files. As indicated in the legend of the heatmap tables, the boxed cells reflect those that reached statistical significance. We have now additionally reflected this in both the legend and the Heatmaps themselves (next to the color scale).

Reviewer #2 (Major Comments for the Author):

In this work, Baktash and colleagues retrospectively investigate an increase in *Clostridioides difficile* ribotype 002 CDI incidence in the Netherlands. Using data obtained from national surveillance, the authors identified a previously unrecognized increase in RT002 CDI incidence between 2017-2018 with a reported decrease in disease severity compared to "hypervirulent" ribotypes. This was not identified from cases isolated prior to the increased incidence (2009-2016). Amongst a total of 8080 ribotypes isolates (2009-2021), 611 RT 002 isolates were identified. Sequencing of 39 randomly selected isolates revealed no relation to isolates obtained from a 2017 RT002 outbreak in Ireland by core genome MLST. While the 12-year retrospective data is interesting, it is difficult to draw meaningful conclusions concerning potential changes in strain epidemiology and linkage to patient response/outcomes from what is reported. This is further confounded by the sequencing of a relative few of the Dutch RT002 isolates. Major and minor comments to follow.

Major comments

1. The use of 39 randomly selected RT002 isolates from a total of 611 (~6%) seems to leave the possibility there could be undetected relatedness both among RT002 isolates in the Netherlands, as well as those from Ireland. It is difficult to draw meaningful conclusions using such a small number of sequenced strains in what appears to be a very heterogeneous sampling, particularly as the authors suggests there may be important clinical differences between isolates from 2009-2017 versus those from 2017-2021. Why were only 39/611 isolates sequenced? Data should be provided concerning when these sequenced strains were isolated, and data from additional Dutch RT002 strains should be considered.

We acknowledge the fact that sequencing a larger number of strains further improves the manuscript; our choice for n=39 strains was motivated by resources available for the study and these prevent us from including further isolates. We agree that it is useful to include metadata on the sequenced isolates; we now have provided Supplemental Table S1 that contains details about all sequenced isolates, which demonstrates that they are geographically and temporally diverse.

Reviewer #2 (Minor Comments for Author (Required)):

1. Lines 16: "...due to clonal RT002 strains..." - remove extra a.

Removed.

Reviewer #3 (Major Comments for the Author):

1. Definition of 'complications' in lines 101-103. Were toxic megacolon and ileus recorded as complications of CDI? Because these are uncommon complications, I don't believe that their absence would invalidate the analysis, but their inclusion would be useful. Also "need for a surgical procedure" is broad - were the authors referring to colectomy as a complication of CDI?

The applied definitions were developed in close collaboration with ECDC to standardize European surveillance amongst participating hospitals to assess its feasibility. Toxic megacolon and ileus are very rare and considered as severe CDI and as complicated CDI when the patient had a surgery

needed or was admitted to the ICU because of CDI. The need for surgery was only recorded when CDI was the reason.

2. Ethics section starting on line 112: were the data de-identified?

All the clinical and microbiological data are pseudo anonymized and cannot be traced back to individual patients.

3. Statistical analysis: in addition to age, did the authors have access to data related to immune status and/or underlying gastrointestinal conditions? If available, these may be useful covariates for the analyses.

Unfortunately, such specific data was not available. Only since 2019, data to assess the severity of the underlying disease using the McCabe score is collected. Therefore, we could not use it to correct for confounders.

4. Lines 184-187 "We compared demographic data...": this material should be in the statistical analysis section.

These lines were removed from the Results section and were already mentioned in the Material & Methods section.

5. Paragraph starting on line 208. The analysis of the characteristics of RT002 prior to and after the increase in prevalence is interesting but I think the issue of treatment needs to be addressed as a possible confounder. Was treatment recorded during data extraction? During the periods analyzed, at least in the US, there was a shift of first line CDI treatment from oral metronidazole to oral vancomycin that could impact outcomes. Did a similar shift in treatment preferences occur in the study population? Also, were there any other changes in the study population or CDI management that occurred during the study periods that should be considered?

We agree that treatment is a potential confounder, but we have not collected data on anti-CDI treatment in the Dutch national surveillance. In line with the response to reviewer one, we have included a statement in the Discussion to acknowledge this limitation.

The national "Stichting Werkgroep Antibioticabeleid (SWAB)" with treatment advice for gastroenteritis suggested metronidazole as the first line treatment for non-severe CDI in 2014 but changed into vancomycine/ fidaxomicine in the guideline from 2023 (see <https://adult.nl.antibiotica.app/en/node/7939>). So, there was no major shift in treatment during our study period.

6. The authors consider biological factors underlying changes in the clinical characteristics of different RTs in the discussion (ex. the paragraph starting on line 310) but they should address clinical factors as per item #5 above.

As per item #5 above, we have included a statement to this effect (Line 338-344).

7. Comparison of Dutch and Irish RT002 strains. This material is interesting, but it feels separate and distinct from the clinical analysis of RT002 compared to other strains. This material may be more appropriate for a separate manuscript.

We appreciate the view of the reviewer; however, the present study was initiated by the report of the Irish outbreak/clonal complex as indicated (and now referenced) in the Introduction to the manuscript. We therefore respectfully disagree, and feel it is an integral part of the present work.

8. Studies in the US have suggested that children with CDI carry strains that are different from adults (see the discussion section of Gomez et al. Diagn Microbiol Infect Dis. 2018. 91(3): 229-232.) An analysis of strains by age group may be interesting, although such an analysis might be out of the scope of this paper.

We agree that differences in CDI epidemiology between age groups is interesting. In fact, we have published on this topic already in 2017 in Clin Infect Dis (PMID: 27986664) and found PCR ribotype 265, primarily in children (< 18 years of age) with lower mortality and complication rates. In this study we found that patients with CDI due to RT001 has a higher median age than RT002 (even after correction for antibiotic usage $P < 0.001$).

Reviewer #3 (Minor Comments for Author (Required)):

None

Reviewer #2 (Comments for the Author):

This manuscript provides a valuable genetic analysis of the epidemiology of *Clostridioides difficile*. The study is strengthened by both an adequate survey period and a sufficient number of isolates analyzed, which together enhance the scientific significance of the presented data. The detection of an increase in ribotype 002 is a particularly important finding and stands out as a key observation in this work. However, the academic value of the manuscript could be further elevated if the authors were able to present data suggesting potential factors underlying this increase. Elucidating such factors would provide deeper insight into the observed epidemiological trends.

We thank the reviewer for the positive evaluation of the quality and soundness of the study. Though we agree that the academic value of the study would increase if we could positively identify the mechanisms underlying our observations, this was not the aim of the study. Rather, we set out to investigate the clinical characteristics associated with RT002 infections, and to determine if the Dutch RT002 isolates were related to Irish isolates.

The identification of an association between RT002 and disease severity is noteworthy, as it highlights an important issue. Nevertheless, the genetic analyses do not yet provide in-depth insights into the pathogenic characteristics of the strains. While the volume of data suggests that the study has potential as a surveillance report, the current manuscript lacks sufficient figures and tables to fully support this approach. As such, readers interested in utilizing the epidemiological data may find the presentation insufficient.

We included 2 figures, 2 tables, 1 supplemental figure and 7 Supplemental Tables as well as publicly available whole genome sequence data referenced in the manuscript. It is unclear to the authors what other information the reviewer would like to see. We have not included negative results (no differences) as we feel this does not contribute to the understanding of the reader and is only cited in the Discussion. However, as an example, we show below (in the comments to reviewer #4) that the PaLoC composition of the isolates is highly similar. We also provide this information for review as *.aln and Excel files.

Fundamentally, this is an epidemiological study enhanced by genetic analysis. In terms of both the title and the overall narrative, the relationship between RT002 and disease severity does not appear to be the main focus, but rather a secondary finding that emerged from the broader analysis. This positioning seems appropriate given the current structure and content of the manuscript.

We appreciate that the reviewer considers the focus of the manuscript appropriate.

Reviewer #3 (Comments for the Author):

Phenotypic antimicrobial resistance results should be included. These results may also be able to characterize RT002 from 2009 to 2017 and RT002 from 2017 to 2021.

As delineated in the Abstract, and acknowledged by Reviewer #2, the goal of this study was not to assess antimicrobial resistance of all isolates. Nevertheless, as indicated in the original manuscript, we have data on susceptibility for a subset of the strains covering both periods (n=28) that reveal no differences in median MIC for metronidazole (MIC=0.125 mg/L) and vancomycin (MIC<0.06 mg/L), the two indicated treatment antibiotics in the study period (fidaxomicin was not used in the Netherlands in that period). We provide this data below, clarified the text and have added the median MIC's to the manuscript.

	metronidazole MIC (mg/L)	vancomycin MIC (mg/L)
RT002-2016-01	0.125	<0.06
RT002-2016-02	0.125	<0.06
RT002-2016-03	0.125	0.06
RT002-2016-04	0.125	0.06
RT002-2016-05	0.125	0.06
RT002-2016-06	0.06	<0.06
RT002-2017-1-a	0.06	<0.06
RT002-2017-2-a	0.06	0.06
RT002-2017-3-a	0.06	<0.06
RT002-2017-4-b	0.125	0.06
RT002-2017-5-b	0.125	0.06
RT002-2017-6-b	0.125	0.06
RT002-2017-7-b	0.125	<0.06
RT002-2017-8-b	0.06	<0.06
RT002-2018-01	0.125	<0.06
RT002-2018-02	0.06	<0.06
RT002-2018-04	0.06	0.06
RT002-2018-05	0.125	0.06
RT002-2018-06	0.125	0.06
RT002-2019-02	0.125	<0.06
RT002-2019-03	0.125	<0.06
RT002-2019-04	0.125	<0.06
RT002-2019-05	0.125	<0.06
RT002-2019-06	0.125	<0.06
RT002-2019-07	0.06	<0.06
RT002-2019-08	0.125	<0.06
RT002-2019-09	0.06	<0.06
RT002-2019-10	0.125	<0.06

Reviewer #4 (Comments for the Author):

This is an interesting retrospective study relating the epidemiology of CDI caused by RT 002 strains in the Netherlands over a very long observational period. The authors analysed a very large dataset comprising patient data from 2009 to 2021. Out of 8080 strains confirmed to be RT 002 by PCR ribotyping, 39 strains were selected at random. These strains were fully sequenced and compared by MLST methods from the obtained genome sequences. These genome sequences have been uploaded to NCBI and will be an asset to the *C. difficile* research community.

The authors identified two distinct time periods in which the earlier period (2009-2017) corresponded with increased CDI severity that was more comparable to hypervirulent strains than strains of normal virulence. In the later period (2017-2021) although the incidence of CDI ascribed to RT 002 in NL was increased, the severity of CDI was more comparable to strains of standard virulence and less comparable to infection caused by hypervirulent ribotypes. This is an interesting observation that could not be explained by analysis of the WGS from strains isolated in the earlier and later periods.

Finally, the RT 002 strains isolated in NL during the surveillance period were shown to be genetically distinct from those isolated during a previous outbreak in Ireland. Clonal relatedness was however observed across the various testing sites in NL.

This study provides insight into infection caused by RT 002 *C. difficile* in NL and the changing epidemiology observed across a long period of observation.

We appreciate the positive evaluation.

Major comments

I have reviewed this paper after the initial round of peer review. From my analysis of the review comments and author responses from the first round, the comments to all 3 reviewers are addressed satisfactorily, in my opinion.

We thank the reviewer for the careful assessment of our manuscript, and the changes implemented after the review by ASM's Journal of Clinical Microbiology. We are glad to learn that reviewer feels the changes we have made were appropriate.

Difference between RT002 in May 2009-April 2017 and May 2017- April 2021 strains. This is very interesting. When discussing that no obvious mutations relating to virulence or sporulation were observed in the conclusions and discussion, please state the number of sequenced strains that correspond to the earlier time period (more severe CDI) and later time period (less severe CDI).

We have added this information (which was contained in Table S1) to the manuscript, as requested.

Line 376-377 - It is not clear from this statement whether any genetic differences were observed between these two groups within the conserved PaLoc region at all. The current statement claims that no mutations associated with virulence were detected. But any mutation in this region could have an impact. Please clarify this point.

We have clarified this point; though minor variations were observed, none were associated exclusively with one group or the other.

For the reviewer's understanding, we show some of the data below. As these data are negative, we choose not to include it in the Discussion section of the manuscript.

The PaLoc, specifically, is highly conserved (clinker image showing homology in the PaLoc between all RT002 isolates):

A clustalOmega alignment indicates that there appear to be two subtypes of PaLoc in the RT002 strains based on nucleotide sequence, but these are distributed among both time periods and therefore cannot explain the difference in virulence. Moreover, both protein variants are still classified as A1.1 and B1.2 by the DiffBase toxin classification (100% identity, Evalued 0). We provide the full clustalOmega data as *.aln file for the purpose of the review and show only the tree below to illustrate this point:

We additionally performed analyses using PubMLST. As expected from the sequence type assignment, we see that all strains are MLST type 8. Looking at allele numbers across a wider panel of virulence-associated (partially non-core genome) loci as available there (e.g. *cwp* and *pbp* genes), we see very limited variation in allele assignment. Similar to the PaLoc, there are no alleles specifically associated with a particular time period, and numbers are too low to infer significance of potential shifts in prevalence of individual alleles. These data are provided as an Excel file for review.

minor comments

Line 66-67 Abbreviations introduced in the abstract (CDI, RT) should be reintroduced here.

We have reintroduced the abbreviations.

Line 122- define EC-PCR Ribotyping

This is capillary electrophoresis PCR ribotyping (CE-RT); this is now written in full.

Line 201 - "We were wondering" is quite informal. Please rephrase more formally e.g. In order to ascertain whether

Rephrased as suggested.

Line 247 - This title is a little confusing at this step of the manuscript. Consider replacing "proportion increase" with the actual dates: CDI in hospitalized patients due to RT002 in May 2009-April 2017 was severe, and comparable with RT027 and RT078/126 strain

Corrected as suggested.

Line 280 - Should this title read "Dutch RT002 strains ARE not related to the Irish outbreak strains, but certain Dutch strains..."

The reviewer is correct, and the change has been made.

Line 376-377 - Please reference source papers relating to mutations in the PaLoc that affect virulence, rather than the selected review.

We appreciate the reviewer's suggestion to include primary papers; however, to include one or more of these, would bias the reader based on the references we choose, as we cannot cite all; we therefore prefer to cite a review that gives a more comprehensive view of the literature describing such mutations.

Re: Spectrum01446-25R1-A (Increase of *Clostridioides difficile* PCR ribotype 002 infection coincides with decreased disease severity in hospitalized patients in the Netherlands)

Dear Dr. Wiep Klaas Smits:

Thank you for the privilege of reviewing your work. Below you will find my comments, instructions from the Spectrum editorial office, and the reviewer comments.

Revision Guidelines

Sincerely,
Yuan Pin Hung
Editor
Microbiology Spectrum

Reviewer #4 (Comments for the Author):

In my opinion, the authors have complied with all reasonable requests from the reviewers at this stage. The additional information provided has improved the clarity of the manuscript as has addressing typographical errors in the text and titles.

The inclusion of median AMS values is a welcomed addition to the manuscript.

Although not included in the manuscript itself, the authors provided the reviewers with multiple analyses to suggest that genetic differences in the PaLoc cannot explain the difference in CDI severity observed across the two distinct time periods. I am satisfied with the evidence presented. I understand the authors' decision not to formally include these data into the manuscript, considering the manuscript already has a large number of figures, supplemental figures and supplemental data files, and considering that the presented data are indeed negative findings.

Reviewer #5 (Comments for the Author):

Dear authors,

Many thanks for the revised version of the manuscript. From my side there are no further alterations necessary.

With the best regards

Reviewer #6 (Comments for the Author):

Comments:

1. The study relies on retrospective data from 2009-2021, which can be affected by incomplete records, reporting bias over time.
 2. While core genome MLST was used for strain comparison, the study does not investigate functional genomic traits (e.g., toxin gene regulation) that may explain phenotypic differences in severity or spread.
 3. The study only includes isolates from the Netherlands and Ireland, which may limit the generalizability of findings to broader European or global trends.
 4. The authors suggest the increased prevalence may be due to clinical practice changes, but no data on treatment regimens, antibiotic stewardship policies were provided.
 5. While genetic clustering of some Dutch RT002 strains was observed, lack of patient-level epidemiological data (e.g., travel history) prevents strong conclusions about possible transmission routes.
- So, it is highly recommended to have a paragraph to discuss the major limitations of the present study before the section of the conclusion.

Minor comments:

Move the key words to be after the abstract not before. Also, rephrase it to be more representative for your study; you added "Clostridioides difficile; Clostridioides difficile infection". The aim from the key words to make your study visible in search, so add something different representing your work.

Response to reviewers

Reviewer #4 (Comments for the Author):

In my opinion, the authors have complied with all reasonable requests from the reviewers at this stage. The additional information provided has improved the clarity of the manuscript as has addressing typographical errors in the text and titles.

The inclusion of median AMS values is a welcomed addition to the manuscript.

Although not included in the manuscript itself, the authors provided the reviewers with multiple analyses to suggest that genetic differences in the PaLoc cannot explain the difference in CDI severity observed across the two distinct time periods. I am satisfied with the evidence presented. I understand the authors' decision not to formally include these data into the manuscript, considering the manuscript already has a large number of figures, supplemental figures and supplemental data files, and considering that the presented data are indeed negative findings.

Response: thank you.

Reviewer #5 (Comments for the Author):

Dear authors,

Many thanks for the revised version of the manuscript. From my side there are no further alterations necessary.

With the best regards

Response: thank you.

Reviewer #6 (Comments for the Author):

Comments:

1. The study relies on retrospective data from 2009-2021, which can be affected by incomplete records, reporting bias over time.
2. While core genome MLST was used for strain comparison, the study does not investigate

functional genomic traits (e.g., toxin gene regulation) that may explain phenotypic differences in severity or spread.

3. The study only includes isolates from the Netherlands and Ireland, which may limit the generalizability of findings to broader European or global trends.

4. The authors suggest the increased prevalence may be due to clinical practice changes, but no data on treatment regimens, antibiotic stewardship policies were provided.

5. While genetic clustering of some Dutch RT002 strains was observed, lack of patient-level epidemiological data (e.g., travel history) prevents strong conclusions about possible transmission routes.

So, it is highly recommended to have a paragraph to discuss the major limitations of the present study before the section of the conclusion.

Response: Several of the points raised by the reviewer were already addressed in the previous version of the manuscript, including a paragraph describing limitations of the study, but their phrasing is adjusted to reflect the reviewer's comments in the new round of revision. Missing points were added to the Limitations paragraph.

Minor comments:

Move the key words to be after the abstract not before. Also, rephrase it to be more representative for your study; you added "Clostridioides difficile; Clostridioides difficile infection". The aim from the key words to make your study visible in search, so add something different representing your work.

We have adapted the keywords; we removed Clostridioides difficile. We kept "Clostridioides difficile infection (CDI)" (as the disease is not mentioned in the title). In addition to "RT002" and "core genome MLST", we now included "incidence", "severe CDI", "complicated CDI", "molecular epidemiology" and "sentinel surveillance".

We also moved them after the Abstract and Importance paragraph (as these sections appear together in the journal. However, we note that the use and position of the keywords is subject to ASM style and we trust that the editorial team takes care of this point should further edits be necessary.

Re: Spectrum01446-25R2 (Increase of *Clostridioides difficile* PCR ribotype 002 infection coincides with decreased disease severity in hospitalized patients in the Netherlands)

Dear Dr. Wiep Klaas Smits:

Your manuscript has been accepted, and I am forwarding it to the ASM production staff for publication. Your paper will first be checked to make sure all elements meet the technical requirements. ASM staff will contact you if anything needs to be revised before copyediting and production can begin. Otherwise, you will be notified when your proofs are ready to be viewed.

Sincerely,
Yuan Pin Hung
Editor
Microbiology Spectrum

Reviewer #4 (Comments for the Author):

I was already satisfied with the improvements made by the authors at the previous round of revisions. I am still satisfied in this most recent round of revision. The nature of changes requested were minor and have been adequately addressed.

Reviewer #5 (Comments for the Author):

As with the last review. I do not have any further questions.

However, since reviewer #6 asked about some epidemiologic details. Maybe in the discussion section the molecular epidemiology in neighbouring countries could be assessed in a few sentences.

Reviewer #6 (Comments for the Author):

Dear authors,
I have no more comments for the manuscript. Good work overall.